# Density Functional Theory of Highly Excited States of Coulomb Systems

**Ágnes Nagy** 

Department of Theoretical Physics, University of Debrecen, H-4002 Debrecen, Hungary; anagy@phys.unideb.hu

**Abstract:** The density functional theory proposed earlier for excited states of Coulomb systems is discussed. The localized Hartree–Fock (LHF) and the Krieger, Li, and Iafrate (KLI) methods combined with correlation are generalized for excited states. Illustrative examples include some highly excited states of Li and Na atoms.

**Keywords:** density functional theory; Coulomb systems; excited states



## 1. Introduction

The density functional theory (DFT) was initially a ground-state scheme [1,2], although in special cases the ground-state constrained-search functional [3] yields exact excited-state energies. Rigorously, DFT was extended to excited states as a subspace theory by Theophilou [4] and later to a more general ensemble theory by Gross, Oliveira and Kohn [5–7]. These approaches have found several extensions and applications (e.g., [8–26]). Nevertheless, they have the unfavorable feature that all levels lying below the excited state intended to study should be taken into account in the calculation. It can be rather inconvenient in the case of higher excited states. Hence, approaches for an individual excited state were worked out.

It was proved that in Coulomb systems the density determines the external potential even in excited states [27–29]. Then, a variational bifunctional theory was put forward [30,31] and analyzed [32,33]. Several other significant schemes were developed [34–36]; however, time-dependent DFT [37–39] is still usually employed for excited states.

In a series of papers [40–42], a comprehensive theory for excited states of Coulomb systems is put forward. It is based on the fact that the Coulomb density determines not only its Hamiltonian but the degree of excitation as well. It makes it possible to develop a universal functional valid for any excited state. Moreover, the excited-state Kohn–Sham (KS) equations are similar to the ground-state KS equations.

Unfortunately, the exact form of the exchange-correlation functional is not known. It is not unexpected, as the exchange-correlation functional is unknown even for the ground state. Therefore, this functional should be approximated in calculations. We mention in passing that several exact constraints that the excited-state functionals should satisfy have recently been derived [43,44].

If we consider the exchange as a functional of the orbitals instead of the density, the energy functional is known: it is the well-known Hartree–Fock expression with the KS orbitals, of course. The exchange potential—which is a local potential in DFT—can be obtained by the optimized potential method (OPM). In the ground-state theory, several methods have been proposed to find the local potential whose eigenfunctions would minimize a given energy functional [45,46]. The localized Hartree–Fock (LHF) [47–49] and the KLI (Krieger, Li, and Iafrate) [50–52] methods proved to be excellent approximations to the OPM. An alternative derivation to the KLI method was also proposed by the present author [53].

Here, extensions of the LHF and KLI methods combined with correlation are proposed to DFT of Coulombic excited states. These approaches provide an almost exact treatment

of exchange and can be coupled with any approximate correlation functional. It is worth starting with a very simple approximate correlation functional, so the local Wigner expression is taken here [54]. As an illustration total and excitation energies are presented for some (including highly) excited states of Li and Na atoms.

The paper is organized as follows. In Section 2, the DFT for Coulombic excited states [40–42] is summarized. Section 3 presents generalization of the LHF and KLI methods combined with correlation. Section 4 is dedicated to the discussion.

## 2. DFT for Coulombic Excited States

First, the theory of Coulombic excited states is summarized. Consider a system in an external Coulomb potential of the form

$$v^{Coul}(\mathbf{r}) = -\sum_{\beta=1}^{M} \frac{Z_\beta}{r_\beta}, \tag{1}$$

where $r_\beta = |\mathbf{r} - \mathbf{R}_\beta|$ and $M$ is the number of nuclei. $\mathbf{R}_\beta$ and $Z_\beta$ stand for the position and the charge of the nucleus $\beta$. The Hamiltonian is

$$\hat{H} = \hat{T} + \hat{V}_{ee} + \sum_{i=1}^{N} v^{Coul}(\mathbf{r}_i), \tag{2}$$

where $\hat{T}$ and $\hat{V}_{ee}$ are the kinetic energy and the electron-electron energy operators. Kato's theorem [55–61]

$$\left. \frac{\partial \bar{n}_\beta(r_\beta)}{\partial r_\beta} \right|_{r_\beta=0} = -2Z_\beta n(\mathbf{r} = \mathbf{R}_\beta) \tag{3}$$

is valid for an excited state, too. Therefore, the cusps of the density $n$ uncover the atomic numbers and the positions of the nuclei. On the other hand, the integral of the density generates the number of electrons. That is, the density provides all parameters of the Coulomb potential (1), consequently determining the external potential, the Hamiltonian (2), and all properties of the Coulomb system. Moreover, a Coulombic electron density cannot be a stationary state density for any other Coulomb external potential, and two different excited states cannot have the same electron density, as proved in [40]. Therefore, the functional

$$F^{Coul}[n] = E[n] - \int n(\mathbf{r}) v^{Coul}[n;\mathbf{r}] d\mathbf{r} \tag{4}$$

can be defined for Coulombic densities. Unfortunately, no easy method is available to decide whether a given density is Coulombic or not; consequently, it is worth defining the functional $F$ for all electron densities.

Consider first a bifunctional

$$F[n, n^{Coul}] = \min_{\substack{\Psi \to n \\ \{\langle \Psi | \Psi_l^{Coul}[n^{Coul}] \rangle = 0\}_{l=1}^{k-1}}} \langle \Psi | \hat{T} + \hat{V}_{ee} | \Psi \rangle. \tag{5}$$

In Equation (5), the minimization is performed with the constraint that each wave function gives the excited-state density $n$ and is orthogonal to the first $k-1$ eigenfunctions of the Coulomb system of $n^{Coul}$. The existence of a Coulomb density close to $n$ is assumed. If more than one Coulomb density can be found at the same distance from $n$, the one with the smallest $F$ is taken.

$$F^{Coul}[n] = F_{\epsilon_{min}}^{Coul}[n], \tag{6}$$

where

$$F_\epsilon^{Coul}[n] = \min_{n^{Coul}} F[n, n^{Coul}]; \quad ||n^{Coul} - n|| \le \epsilon. \tag{7}$$

The best measure for the distance is not known yet. We prefer the Sobolev-type norm:

$$d(n^{Coul}, n) \equiv \int \left| \sqrt{n^{Coul}(\mathbf{r}) - n(\mathbf{r})} \right|^2 d\mathbf{r} + \int \left| \nabla \sqrt{n^{Coul}(\mathbf{r}) - n(\mathbf{r})} \right|^2 d\mathbf{r}. \tag{8}$$

The mathematical properties of the functional might depend on the definition. Therefore, this problem should be the subject of future investigation (see further details in [40]). The minimization leads to the Euler equation

$$v^{Coul}([n], \mathbf{r}) = -\frac{\delta F^{Coul}[n]}{\delta n(\mathbf{r})} \tag{9}$$

up to a constant.

In calculations, the Kohn–Sham (KS) system is preferred; therefore, it is valuable to define the non-interacting kinetic energy

$$T_s[n^{Coul}] = \min_{\substack{\Phi \to n^{Coul} \\ \{\langle \Phi | \Phi_j[n^{Coul}] \rangle = 0\}_{j=1}^{l-1} \\ ||n_1^{Coul} - n_1^0|| \le \delta}} \langle \Phi | \hat{T} | \Phi \rangle. \tag{10}$$

The minimization is over the wave functions $\Phi$ having the excited-state density $n^{Coul}$ and orthogonal to the first $l-1$ eigenfunctions of the non-interacting system. That is, $n^{Coul}$ is the same in the real and the KS systems. However, the ground states can be different. It may happen that there are more than one KS system with the density $n^{Coul}$. Then, take the one in which the KS ground-state density $n_1^0$ is closest to the true ground-state density $n_1^{Coul}$.

We need the KS kinetic energy as a functional of a not necessarily Coulomb density $n$. First, a bifunctional is defined:

$$T_s^{Coul}[n, n^{Coul}] = \min_{\substack{\Phi \to n \\ \{\langle \Phi | \Phi_l[n^{Coul}] \rangle = 0\}_{l=1}^{k-1} \\ ||n_1^{Coul} - n_1^0|| \le \delta}} \langle \Phi | \hat{T} | \Phi \rangle. \tag{11}$$

The minimum is taken with the constraint that the density of the $l$th KS state $\Phi$ equals $n$, $\Phi$ is orthogonal to all states $\Phi_j$, $1 \le j < l$ and the ground-state KS density $n_1^0$ is as close as possible to the real ground-state density $n_1^{Coul}$. We assume the existence of a unique Coulomb density close to the non-Coulomb density $n$ and construct the functional

$$T_{s,\epsilon}^{Coul}[n] = \min_{n^{Coul}} T_s^{Coul}[n, n^{Coul}], \tag{12}$$

where

$$||n^{Coul} - n|| \le \epsilon. \tag{13}$$

It is expected that there is at least one Coulomb density closer to $n$ than $\epsilon$, when taking a large enough value for $\epsilon$. Finally, the smallest value of $\epsilon$ provides the kinetic energy functional

$$T_s^{Coul}[n] = T_{s,\epsilon_{min}}^{Coul}[n]. \tag{14}$$

The variational principle provides an Euler equation, within an additive constant,

$$w^{Coul}([n], \mathbf{r}) = -\frac{\delta T_s^{Coul}[n]}{\delta n(\mathbf{r})}. \tag{15}$$

To derive the KS equations it is helpful to partition $F^{Coul}[n]$ as

$$F^{Coul}[n] = T_s^{Coul}[n] + J^{Coul}[n] + E_{xc}^{Coul}[n], \tag{16}$$

where $J^{Coul}[n]$ and $E_{xc}^{Coul}[n]$ are the classical Coulomb and exchange-correlation energies. Comparing Equations (9), (15), and (16), the KS potential

$$w^{Coul}([n], \mathbf{r}) = v^{Coul}([n], \mathbf{r}) + v_J^{Coul}([n], \mathbf{r}) + v_{xc}^{Coul}([n], \mathbf{r}) \tag{17}$$

is obtained as the sum of the external, the classical Coulomb and the exchange-correlation potentials. The KS equations have the form

$$\left[ -\frac{1}{2}\nabla^2 + w^{Coul}([n], \mathbf{r}) \right] \phi_i = \varepsilon_i \phi_i, \tag{18}$$

where the KS orbitals $\phi_i$ provide the density as

$$n = \sum_{i=1}^{K} \lambda_i |\phi_i|^2, \tag{19}$$

where the occupation numbers $\lambda_i$ are 0, 1, or 2 for a non-degenerate system. $K$ stands for the orbital having the highest orbital energy with non-zero occupation number.

## 3. Orbital-Dependent Exchange-Correlation Functional

The exact form of the exchange-correlation functional is unknown and has to be approximated in calculations even in the original ground-state DFT. As the theory for Coulombic excited states studied in this paper is also valid for the ground state, it is worth testing how the ground-state functionals work. The exchange is known exactly as a functional of orbitals. This functional is the Hartree–Fock (HF) expression. The Kohn–Sham potential is local, therefore a local exchange potential should be generated. In the exchange-only approximation, the so-called optimized potential method (OPM) [45,46] or the localized Hartree–Fock method (LHF) [47–49] can be applied. In the latter, it is supposed that the HF and the exchange-only KS determinants are equal. The method have advantageous properties: invariant with respect to unitary transformations of orbitals, the local KS exchange potential is free of self-interaction and, consequently, has correct long-range behavior. Often, instead of OPM or LHF, the KLI (Krieger, Li, and Iafrate) [50–52] approximation is applied. KLI can also be obtained by neglecting certain terms from the LHF exchange potential. In KLI, the exchange potential exhibits the correct long-range behavior, but it is not invariant with respect to unitary transformations of orbitals. KLI is much simpler than OPM and more stable if finite-basis-set is applied. Before LHF approach appeared, the present author also provided an alternative derivation of the KLI method. Now, this method is utilized to extend the LHF and the KLI methods to include correlation in the frame of our excited-state theory.

First, the generalized LHF is derived. Denote it LHFC, where the last letter stands for correlation. The total energy can be written as a functional of orbitals:

$$E^{Coul}[\tilde{\psi}_1, ..., \tilde{\psi}_N] = T_s^{Coul}[\tilde{\psi}_1, ..., \tilde{\psi}_N] + \int n(\mathbf{r})v^{Coul}(\mathbf{r})d\mathbf{r} + J^{Coul}[\tilde{\psi}_1, ..., \tilde{\psi}_N] + E_x^{Coul}[\tilde{\psi}_1, ..., \tilde{\psi}_N] + E_c^{Coul}[\tilde{\psi}_1, ..., \tilde{\psi}_N], \tag{20}$$

where

$$T_s^{Coul} = -\frac{1}{2}\sum_{i=1}^{K} \lambda_i \int \tilde{\psi}_i^*(\mathbf{x})\nabla^2\tilde{\psi}_i\mathbf{x})d\mathbf{x}, \tag{21}$$

$$E_x^{Coul} = -\frac{1}{2}\sum_{i=1}^{K}\sum_{j=1}^{K}\lambda_i\lambda_j\int \tilde{\psi}_i^*(\mathbf{x}_1)\tilde{\psi}_j^*(\mathbf{x}_2)\tilde{\psi}_i(\mathbf{x}_2)\tilde{\psi}_j(\mathbf{x}_1)\frac{1}{r_{12}}d\mathbf{x}_1 d\mathbf{x}_2. \tag{22}$$

$r_{12} = |\mathbf{r}_1 - \mathbf{r}_2|$ and $\mathbf{x} : \mathbf{r}, s$ stand for the spatial and spin coordinates, respectively. The variation of $E^{Coul}$ with respect to the orbitals leads to the equations

$$\left[-\frac{1}{2}\nabla^2 + \hat{u}^{Coul}\right]\tilde{\psi}_i = \sum_{j=1}^{K}\tilde{\epsilon}_{ij}\tilde{\psi}_j, \tag{23}$$

where

$$\hat{u}^{Coul} = v^{Coul} + v_J^{Coul} + \hat{v}_x^{Coul} + \hat{v}_c^{Coul}. \tag{24}$$

$v_J^{Coul}$ is the Coulomb external potential and

$$v_J^{Coul}(\mathbf{r}_1) = \int \frac{n(\mathbf{r}_2)}{r_{12}}d\mathbf{r}_2 \tag{25}$$

is the classical Coulomb potential. $\hat{v}_x^{Coul}$ is a Hartree–Fock-like exchange operator

$$\hat{v}_x^{Coul}\tilde{\psi}_i(\mathbf{x}_1) = -\sum_{j=1}^{K}\lambda_j\int \tilde{\psi}_j^*(\mathbf{x}_2)\tilde{\psi}_j(\mathbf{x}_1)\frac{1}{r_{12}}\tilde{\psi}_i(\mathbf{x}_2)d\mathbf{x}_2 . \tag{26}$$

The form of the correlation potential $\hat{v}_c^{Coul}$ can be obtained from the functional derivative of $E_c^{Coul}$. The correlation functional is unknown; $E_c^{Coul}$ should be approximated. In this paper, a simple local approximation is applied, although the procedure described here can be used for any kind of approximation of the correlation. The right-hand side of Equation (23) arise from the orthonormalization conditions

$$\int \tilde{\psi}_i^*(\mathbf{x})\tilde{\psi}_j(\mathbf{x})d\mathbf{x} = \delta_{ij}, \tag{27}$$

where $\tilde{\epsilon}_{ij}$ are the Lagrange multipliers for the constraint (27).

Consider now the corresponding KS equations arising from the variation of $E^{Coul}$. In doing this, one must actually constrain the orbitals to orthonormal. Then, we arrive at

$$\left[-\frac{1}{2}\nabla^2 + w^{Coul}([n],\mathbf{r})\right]\tilde{\phi}_i = \sum_{j=1}^{K}\tilde{\epsilon}_{ij}\tilde{\phi}_j. \tag{28}$$

Now, we can compare Equation (28) with the correlated Hartree–Fock-like Equation (23). Multiplying Equation (28) by $\lambda_i\tilde{\phi}_i^*$ and Equation (23) by $\lambda_i\tilde{\psi}_i^*$, then summing for all occupied $i$, taking the difference of these equations, and finally using the approximation $\tilde{\psi}_i \approx \tilde{\phi}_i$, we arrive at the exchange-correlation potential of the LHFC approach

$$v_{xcLHFC}^{Coul} = v_{LHFC}^S + v_{LHFC}^{Sc} + v_{LHFC}^e, \tag{29}$$

where

$$v_{LHFC}^S(\mathbf{r}) = \frac{1}{n(\mathbf{r})}\sum_{i=1}^{K}\lambda_i\tilde{\phi}_i^*(\mathbf{x})\hat{v}_x^{Coul}\tilde{\phi}_i(\mathbf{x}) \tag{30}$$

is the Slater potential,

$$v_{LHFC}^{Sc} = \frac{1}{n}\sum_{i=1}^{K}\lambda_i\tilde{\phi}_i^*(\mathbf{x})\hat{v}_c^{Coul}\tilde{\phi}_i(\mathbf{x}) \tag{31}$$

is a Slater-like potential originating from the correlation potential, and

$$v^e_{LHFC} = \frac{1}{n} \sum_{i=1}^{K} \lambda_i \sum_{j=1}^{K} (\tilde{\varepsilon}_{ij} - \tilde{\epsilon}_{ij}) \tilde{\phi}_i^* \tilde{\phi}_j. \tag{32}$$

The last term $v^e_{LHFC}$ appears because the KS and correlated Hartree–Fock-like equations have different Lagrange multipliers. Using Equations (28) and (23), the difference $\tilde{\varepsilon}_{ij} - \tilde{\epsilon}_{ij}$ can be expressed as

$$\tilde{\varepsilon}_{ij} - \tilde{\epsilon}_{ij} = \langle \tilde{\phi}_i | v^{Coul}_{xc} - \hat{v}^{Coul}_x - \hat{v}^{Coul}_c | \tilde{\phi}_j \rangle. \tag{33}$$

Equation (29) provides the exchange-correlation potential. Observe that neglecting the correlation gives the LHF exchange potential of Della-Sala and Görling [47]. If all terms $i \neq j$ are neglected in Equation (32), we arrive at the KLI method with correlation. Omitting correlation, the original KLI approach is given.

The KLI and KLIC potentials can be derived explicitly from the canonical forms of Equations (23) and (28). Equations (23) and (28) can be reformalized by unitary transformations of orbitals:

$$\left[ -\frac{1}{2} \nabla^2 + \hat{u}^{Coul} \right] \psi_i = \epsilon_i \psi_i, \tag{34}$$

and

$$\left[ -\frac{1}{2} \nabla^2 + w^{Coul} \right] \phi_i = \varepsilon_i \phi_i, \tag{35}$$

respectively. $\hat{u}^{Coul}$ and $w^{Coul}$ have the same form as before, but are expressed with the canonical orbitals $\psi_i$ and $\phi_i$ instead of $\tilde{\psi}_i$ and $\tilde{\phi}_i$. Following the same steps we did in the derivation of Equation (29), we arrive at the KLI-like exchange-correlation potential

$$v^{Coul}_{xcKLIC} = v^{S}_{KLIC} + v^{Sc}_{KLIC} + v^{e}_{KLIC}, \tag{36}$$

where

$$v^{S}_{KLIC}(\mathbf{r}) = \frac{1}{n(\mathbf{r})} \sum_{i=1}^{K} \lambda_i \phi_i^*(\mathbf{x}) \hat{v}^{Coul}_x \phi_i(\mathbf{x}) \tag{37}$$

is the Slater potential,

$$v^{Sc}_{KLIC} = \frac{1}{n} \sum_{i=1}^{K} \lambda_i \phi_i^*(\mathbf{x}) \hat{v}^{Coul}_c \phi_i(\mathbf{x}) \tag{38}$$

is a Slater-like potential originating from the correlation potential, and

$$v^e_{KLIC} = \frac{1}{n} \sum_{i=1}^{K} \lambda_i (\varepsilon_i - \epsilon_i) |\phi_i|^2. \tag{39}$$

The last term $v^e_{KLIC}$ appears because the KS and correlated Hartree–Fock-like equations have different one-electron energies. Using Equations (35) and (34), the difference $\varepsilon_i - \epsilon_i$ can be expressed as

$$\varepsilon_i - \epsilon_i = \langle \phi_i | v^{Coul}_{xc} - \hat{v}^{Coul}_x - \hat{v}^{Coul}_c | \phi_i \rangle. \tag{40}$$

The potential (36) reduces to the KLI exchange potential if the correlation term is omitted. Observe that, even though the same procedure is used to derive both the LHFC

and the KLIC methods, different exchange-correlation potentials are obtained. The reason is that the Lagrange multipliers $\tilde{\varepsilon}_{ij}$ and $\tilde{e}_{ij}$ are different. Therefore, the unitary transformations are also different. Consequently, even the same procedure leads to different approximation.

Now, the methods derived above are illustrated for some excited states of Li and Na atoms. It is natural to start with a very simple approximation for correlation: KLI is combined with the local Wigner approximation [54].

The correlation energy is given by

$$E_c^{LW}[n] = \int \frac{an}{b + r_s} d\mathbf{r} \, , \tag{41}$$

where $r_s$ is the Wigner–Seitz radius:

$$r_s = (3/4\pi n)^{1/3}. \tag{42}$$

The parameters obtained by Süle and Nagy [62] are used: $a = -0.02728$ and $b = 0.21882$. The correlation potential obtained by functional derivation of (41)

$$v_c^{LW} = \frac{a(b + \frac{4}{3}r_s)}{(b + r_s)^2} \tag{43}$$

should be substituted into Equations (38) and (40). Observe that the correlation is taken into account self-consistently.

Table 1 presents the total energies for the ground state and the excited states with configuration $1s^2ms$, $m = 2, ..., 7$ for the Li atom. The KLI values can be directly compared with the exchange-only results of the spin-dependent localized Hartree–Fock (SLHF) [63], xCOEP [64], and Hartree–Fock (HF) [65] methods. x-COEP refers to exchange-only constrained optimized effective potential (xCOEP) methodology [64]. SLHFc is localized Hartree–Fock combined with Lee–Yang–Parr (LYP) correlation [66]. WFLYP [67] refers to work-function-based exchange [68,69] and LYP correlation potentials. Exact energies obtained with accurate configuration interaction wave function in Hylleraas basis set [70] are presented in the last column. The KLI, SLHF, and xCOEP values are very close to the HF results, the SLHF values are closest. We can see that KLI approximates SLHF and HF excellently. The KLI with local Wigner correlation (KLILW) leads to lower total energies than the exact ones. SLHFc and WFLYP give more accurate total energies.

**Table 1.** Total energies in Rydberg units for the ground and some excited states of the Li atom.

| State | KLI | KLILW | SLHF | SLHFc | WFLYP | xCOEP | HF | Exact |
|-------|-----|-------|------|-------|-------|-------|-----|-------|
| $1s^22s$ | −14.8643 | −14.9960 | −14.8650 | −14.9744 | | −14.8634 | −14.8655 | −14.9561 |
| $1s^23s$ | −14.6198 | −14.7450 | −14.6201 | −14.7191 | −14.7155 | −14.6146 | −14.6204 | −14.7082 |
| $1s^24s$ | −14.5494 | −14.6726 | −14.5496 | −14.6463 | −14.6396 | −14.5463 | −14.5498 | −14.6371 |
| $1s^25s$ | −14.5197 | −14.6421 | −14.5198 | −14.6157 | −14.6093 | −14.5145 | −14.5200 | −14.6071 |
| $1s^26s$ | −14.5045 | −14.6264 | −14.5045 | −14.6000 | | −14.4984 | −14.5046 | −14.5917 |
| $1s^27s$ | −14.4956 | −14.6173 | −14.4957 | −14.5909 | | −14.4857 | −14.4957 | −14.5828 |

Calculated and experimental [71] excitation energies of the Li atom are shown in Table 2. The KLI, SLHF, and HF values are very close to each other, while the KLI is a bit worse. While the HF total energies are closer to the exact ones than the xCOEP values, for the excitation energies, xCOEP results are much closer to the exact ones than for the HF data. The KLI values are slightly worse than the HF results, while KLILW and SLHFc overestimate the excitation energies; KLILW values are somewhat better. The first excitation energy computed with time-dependent TDF with ALDA exchange-correlation [72] is less accurate.

**Table 2.** Calculated and experimental excitation energies of the Li atom in Rydberg units.

| State | KLI | KLILW | SLHF | SLHFc | xCOEP | HF | TDDFT | Exact | Exp |
|-------|-----|-------|------|-------|-------|-----|-------|-------|-----|
| $1s^2 3s$ | 0.2445 | 0.2510 | 0.2449 | 0.2553 | 0.2488 | 0.2450 | 0.2280 | 0.2479 | 0.2479 |
| $1s^2 4s$ | 0.3149 | 0.3234 | 0.3154 | 0.3281 | 0.3172 | 0.3157 | | 0.3191 | 0.3191 |
| $1s^2 5s$ | 0.3446 | 0.3539 | 0.3452 | 0.3587 | 0.3489 | 0.3455 | | 0.3490 | 0.3490 |
| $1s^2 6s$ | 0.3598 | 0.3696 | 0.3605 | 0.3744 | 0.3651 | 0.3608 | | 0.3644 | 0.3644 |
| $1s^2 7s$ | 0.3686 | 0.3787 | 0.3693 | 0.3835 | 0.3778 | 0.3697 | | 0.3733 | 0.3733 |

Table 3 presents total energies for the ground and the excited states of the Na atom with configuration $[Ne]ms$, $m = 3, ..., 7$. The KLI and ELP values are very close to the HF results, the ELP ones being a bit closer. ELP (effective local potential) [73] denotes an alternative way of solving the exact-exchange OPM. NHF stands for highly accurate numerical Hartree–Fock method [74]. The KLILW leads to lower total energies than the exact ones.

**Table 3.** Total energies in Rydberg units for the ground and some excited states of the Na atom.

| State | KLI | KLILW | ELP | HF | NHF | Exact |
|-------|-----|-------|-----|-----|-----|-------|
| $[Ne]3s$ | $-323.7106$ | $-324.5443$ | $-323.7116$ | $-323.7174$ | $-323.7178$ | $-324.5092$ |
| $[Ne]4s$ | $-323.4879$ | $-324.3153$ | $-323.4894$ | $-323.4938$ | | |
| $[Ne]5s$ | $-323.4224$ | $-324.2479$ | $-323.4232$ | $-323.4376$ | | |
| $[Ne]6s$ | $-323.3944$ | $-324.2190$ | | | | |
| $[Ne]7s$ | $-323.3798$ | $-324.2040$ | | | | |

Table 4 displays calculated and experimental [71] excitation energies of the Na atom. The KLI, ELP, and HF values are very close to each other. The KLI with local Wigner correlation approximates experimental results quite well.

**Table 4.** Calculated and experimental excitation energies of the Na atom in Rydberg units.

| State | KLI | KLILW | ELP | HF | Exp |
|-------|-----|-------|-----|-----|-----|
| $[Ne]4s$ | 0.2228 | 0.2290 | 0.2222 | 0.2236 | 0.2346 |
| $[Ne]5s$ | 0.2883 | 0.2964 | 0.2884 | 0.2898 | 0.3025 |
| $[Ne]6s$ | 0.3163 | 0.3254 | | | 0.3315 |
| $[Ne]7s$ | 0.3308 | 0.3403 | | | 0.3464 |

It can be concluded that KLI method provides results very close to the HF ones, while the KLILW leads to too low total energies. KLILW overestimates the excitation energies for the Li atom, but it gives much better excitation energies for the Na atom than the KLI or HF method. Of course, these illustrative examples cannot give us full knowledge of the efficiency of the KLILW method, and further studies are necessary. Investigation of other methods and correlation functionals (e.g., [75,76]) will be the subject of future research.

## 4. Discussion

The great advantage of our method is that a single functional is relevant for any bound (ground or excited) state of a Coulomb system. However, of course, we do not know this functional and its properties. It might happen that it is a jagged, discontinuous functional. It can appear if very similar Coulomb densities have very different values of $F^{Coul}$ with the consequence that $F^{Coul}$ would be discontinuous. In our approach, the excitation level of two densities may be different. If two Coulomb densities are close together, they still can have vastly different excitation levels. Therefore, $F^{Coul}$ might be discontinuous. This problem can be avoided by defining functionals $F_k^{Coul}$, that is, using different functionals for different excitation levels $k$. Discontinuities in $F_k^{Coul}$ are much less likely because of the additional dependence on the level of excitation (see further details in [40]). In this paper,

it is supposed that the single functional $F^{Coul}$ is well-behaved and approximations for it are proposed.

We mention in passing that, even though calculations are generally performed in the Kohn–Sham scheme, the Euler equation (9) can also be applied provided that appropriate approximation for the kinetic energy functional is available. Unfortunately, such an approximation is not accessible. However, the Euler equation is very useful. For example, the existence of excited-state Euler equations for specific Shannon information and Fisher information has recently been proved [77]. Even the Euler equation for the relatively specific Shannon information has been derived. It is interesting to note that the Ghosh–Berkowitz–Parr thermodynamic transcription [78] has been recently extended to excited states of Coulomb systems [79]. For Coulomb systems, there is a simple relation between the total energy and phase-space Fisher information both in the ground and excited states. Furthermore, relations for the phase-space fidelity, relative entropy, fidelity susceptibility, and Fisher information have been presented. These kinds of analysis of excited states seem to be important, as excited-state reactivity is a new frontier [80–82].

In summary, Coulombic excited states are studied within the density functional theory proposed earlier. Generalizations of the LHF and KLI methods combined with correlation are derived within DFT for Coulombic excited states. In these approaches, exchange is treated (almost) exactly and any approximate correlation functional can be incorporated. As an illustration, total and excitation energies are presented for some (including highly) excited states of Li and Na atoms.

**Funding:** This research was supported by the National Research, Development and Innovation Fund of Hungary, financed under 123988 funding scheme.

**Institutional Review Board Statement:** Not applicable.

**Informed Consent Statement:** Not applicable.

**Data Availability Statement:** Data sharring is not applicable to this article.

**Conflicts of Interest:** The authors declare no conflict of interest.

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
