# Peer review of "Density Functional Theory of Highly Excited States of Coulomb Systems"

_computation, doi:10.3390/computation9060073_

Round 1

Reviewer 1 Report

The manuscript presents an application of the KLI approximation with the local Wigner correlation in the context of density functional theory for Coulombic excited states. A thorough background of the theory is provided and given for the use of the KLI approximation in this new context. The method is used to calculate total and excitation energies for high energy valence S to S excitations for lithium and sodium atoms. These calculations show that the KLI + local Wigner correlation is a good approximation to the exact energies.

The paper is well written, and the methods presented appear sound, I therefore recommend immediate publication in Computation.

At the authors discretion, it may be nice to include a brief discussion of how the existence of a jagged, discontinuous universal functional may manifest in practical terms. These matters are covered in Ref [40], but a summary may be of interest to unfamiliar readers.

There is a typo of “te” -> “to” on line 161.

Reviewer 2 Report

Density-functional theory (DFT) was originally developed as a ground-state (GS) theory.  Ironically one of the original important applications of GS DFT was as an excited-state (ES) methodology to calculate photoelectron spectra using Slater's transiton-state method.  Even today the lowest ES of a given symmetry is often treated with GS DFT.  Often the  Ziegler-Rauk-Baerends method (or something similar) is applied to extract information about open-shell singlet states from GS DFT and similar GS DFT methods are used in
combination with broken symmetry methods to study the plethera of magnetic states which arise in open-shell systems with more than one metal atom.  All of these approaches use GS DFT as if it were a true ES DFT --- that is, as if the energy were a functional of the ES density --- showing the desirability of (or at least the wide interest in) developing a rigorous ES DFT formalism and practical functionals.  Any such ES DFT must overcome a number of difficulties, including but not limited to the problem of maintaining
orthogonality to all lower energy states and being competitive with the current DFT workhorse for treating excited states, namely excited-state information (including ES densities) obtained via time-dependent (TD) DFT using the response theory formalism.

*** The results reported in the present paper should be compared with

*** those obtained from TD-DFT.

The present manuscript does two more-or-less orthogonal things --- namely

(1) discusses the formal justification of ES DFT, and

(2) calculates the ES energies for Li and Na for ns -> ms excitations via (up to a few nuances) Hartree-Fock theory plus the old Wigner correlation functional.

Only (2) appears in the abstract.

*** Either (1) should also be added to the abstract or it should be removed

*** from the paper.

Regarding (2), the author is solving the optimized effective potential (OEP) problem within the Krieger-Li-Iafrate (KLI) approximation.  The normal OEP problem is equivalent to asking what local potential will product orbitals minimizing the Hartree-Fock energy expression.  In practice OEP energies are very close to Hartree-Fock energies, hence my description of this paper as using "Hartree-Fock theory plus the old Wigner correlation functional." In this paper, the local potential is sought which makes the Hartree-Fock energy expression stationary rather than a minimum since ESs are sought.

*** One thing that must be clarified before this paper is acceptable is

*** whether the Wigner functional is part of the energy expression used

*** to find the noninteracting potential or whether the Wigner functional

*** is only used perturbatively with orbitals determined from the

*** exchange-only theory.

The OEP problem is a numerically ill-conditioned problem.  This problem is often solved by using the KLI approximation which is numerically much better behaved.  However, it has been known since the beginning of the current millenium that the KLI approximation depends upon the initial guess because it fails to invariant under a unitary transformation of the occupied orbitals.  This fatal flaw has been noticed by several workers who have provided the same unitarily invariant correction of the formalism, albeit under several different names:

Localized Hartree-Fock (LHF)
F. Dela Salla and A. Görling, "Efficient localized Hartree–Fock methods as effective exact-exchange Kohn–Sham methods for molecules", J. Chem. Phys. 115, 5718 (2001).

Common Energy Denominator (CEDA)
O. Gritsenko and E.J. Baerends, "Exchange kernel of density functional response theory from the common energy denominator approximation (CEDA) for the Kohn–Sham Green's function", Research on Chemical Intermediates 30, 87 (2004).

Effective Local Potential (ELP)
V.N. Staroverov, G.E. Scuseria, and E.R. Davidson, "Effective local potentials for
orbital-dependent density functionals", J. Chem. Phys. 125, 081104 (2006).

*** The author should cite these works and explain why she is using the

*** ill-defined KLI approximation when she could be using a

*** well-defined approximation.

As this report may seem very negative, let me explain that I find the paper to be creative and I would like to encourage the author in her endeavor to create a practical, yet formally well-justified, ES DFT.  As it stands, I find it too early to recommend publishing this manuscript.  However it may be acceptable after major rewriting to make the objectives and conclusions of the paper much clearer and more focused.  In particular, it must be made much clearer whether this manuscript is a contribution to the formal theory or a computational
test of an existent theory.  In the later case, the formal theory being tested must be made clearer.  Also, as it stands, there is not enough detail for anyone else to be able to reproduce the calculations.  So the missing details need to be provided.  Other criticisms that I have been made could possibly be met with appropriate citations and rewriting.  But it seems to me that a major rewrite is called for before it is worth the effort of this reviewer to consider
this manuscript further.

COMMENTS AND QUESTIONS:

1) Page 2, line 67, "If nore than one Coulomb densities can be found at the same distance from n, then ..." How is "distance" defined?  [i.e., What is the measure used in Eq. (7)?]
2) Page 3, Eq. (8), n should be n({\bf r}).  If {\bf r} appears oh the LHS of the equation, then it should also appear on the RHS of the equation.
3) Same comment for Eq. (14) on page 3.
4) Page 3, Eq. (11), on the LHS of the equation, should \epsilon be \epsilon_{min} ?
5) There is an OEP theory that includes electron correlation

[C95] M.E. Casida, "Generalization of the Optimized Effective Potential Model to Include Electron Correlation: A Variational Derivation of the Sham--Schluter Equation for the Exact Exchange-Correlation Potential'', Phys. Rev. A 51, 2005 (1995).

[GHIB02] I. Grabowski, S. Hirata, S. Ivanov, and R.J. Bartlett, "Ab initio density
functional theory: OEP-MBPT(2). A new orbital-dependent correlation functional", J. Chem. Phys. 116, 4415 (2002).

Why was this not used?

ENGLISH:

1) Page 1, line 30, "well-known"
2) Page 2, line 47, "in an external"
3) Page 2, line 55, "consequently determining the"
4) Page 2, line 61, "worth while to define" or "worth defining"
5) Page 2, line 67, "If nore than one Coulomb density can be found"
6) Page 3, line 77, the grammar is tricky here: "It may happen that there is more than one KS system with"
7) Page 3, line 87, "If more than one Coulomb density can be found"
